# Visual Semantics Meets Medical Diagnosis: Cross-Scale Embedding Alignment for Clinically Explainable Medical Image Segmentation

## Abstract

Medical image segmentation requires explainable AI for clinical deployment, yet Vision-language models like MedSAM [Ma et al., 2024] operate as black boxes. Existing methods like Grad-CAM [Selvaraju et al., 2017] suffer from computational instability and fail to capture multi-modal feature interactions. We present a gradient-free framework generating anatomically-aligned saliency maps across embedding layers via calculated similarity between image features and reference representations. Our three-level methodology progresses from derived insights from image embeddings to organ prototype similarity, prompt-spatial embeddings to a four-component spatial system. Evaluated on CHAOS [Kavur et al., 2021] and FLARE22 [Ma et al., 2023] datasets (13 organs), our approach reveals progressive reasoning: early layers show broad attention, intermediate layers narrow to organ-specific regions, and final layers produce precise boundary identification, enabling clinicians to verify model decisions against medical expertise.

## 1 Introduction

Explainable AI (XAI) in medical imaging addresses the critical gap between high-performing deep learning models and their clinical acceptance [Bhati et al., 2024, Gipiškis et al., 2024]. Although Vision-language models achieve high accuracy in segmenting anatomical structures, their black-box nature prevents clinical adoption, as clinicians cannot verify which image regions drive the decisions.

**Challenges:** Current explainability methods face three key limitations. Gradient-based techniques like Grad-CAM [Selvaraju et al., 2017] are prone to vanishing gradients and computational instability [Suara et al., 2023]. They also produce coarse spatial localization, which is inadequate for precise anatomical verification. Furthermore, they fail to capture the multi-modal interactions between vision encoders and prompt embeddings in modern architectures like MedSAM [Ma et al., 2024].

**Goal:** We aim to develop a gradient-free explainability framework that reveals how Vision-language models build reasoning across layers, generating clinically interpretable saliency maps that align with anatomical structures and capture multi-modal interactions in prompt-based segmentation models.

## 2 Methodology

We implement adaptive contrast enhancement tailored for low-contrast CT and MRI images [Ma et al., 2023]. Our pipeline applies percentile-based stretching and CLAHE with histogram clipping to prevent noise amplification, selectively enhancing foreground anatomical structures while preserving natural background appearance.

Submitted to 39th Conference on Neural Information Processing Systems (NeurIPS 2025). Do not distribute.

The architecture of the segmentation model is built on (1) an image encoder processing inputs into feature maps, (2) a prompt encoder converting bounding box coordinates into spatial embeddings, and (3) a mask decoder generating segmentation outputs. Our framework generates anatomically grounded saliency maps across progressive embedding layers by computing normalized dot products between image features and multiple reference representations at different architectural depths. stage 1 produces diffused maps from visual similarity to organ prototypes (averaged features within bounding boxes), capturing broad anatomical context. stage 2 integrates spatial prompt embeddings with weighted combinations between organ prototypes and box embeddings, narrowing focus to specific structures. stage 3 implements a four-component system combining global image features, organ-specific prototypes, spatial prompt embeddings, and baseline context, with spatial weighting emphasizing regions near bounding box centers while maintaining awareness of distant anatomical context. All similarity scores are normalized and visualized as heatmaps, enabling layer-by-layer analysis of attention progression from coarse to fine-grained anatomical localization.

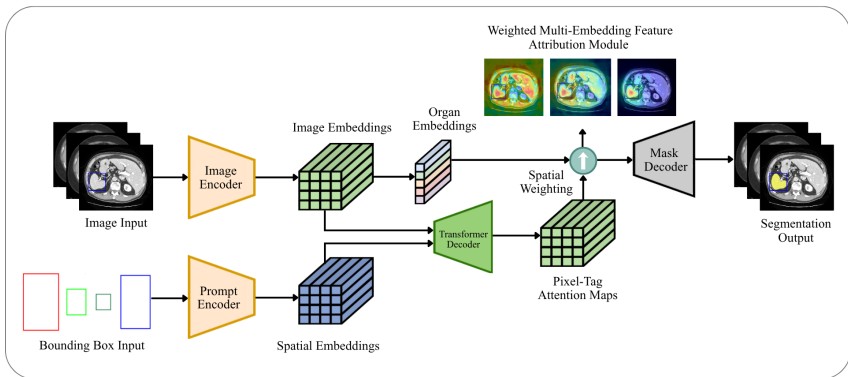

Figure 1: Overview of the multi-embedding explainability framework combining image and prompt encoders with spatial weighting to generate anatomically grounded, layer-wise saliency maps.

# 3 Discussions and Results

We evaluated our framework on CHAOS [Kavur et al., 2021] and FLARE22 [Ma et al., 2023] datasets (50 abdominal CT scans, 13 organs, 100% processing success). The results in Figs. 2-4 show clear progressive refinement across embedding layers: early layers showed broad anatomical attention, intermediate layers narrowed to organ-specific regions, and final layers produced sharp localization on segmentation regions. Sample outputs reveal more localized anatomical alignment compared to Grad-CAM baselines [Selvaraju et al., 2017, Suara et al., 2023], with stable explanations free from gradient-induced noise. The multi-component system successfully concentrated final-layer attention on target organ boundaries while maintaining contextual awareness.

Our gradient-free framework overcomes key XAI limitations by eliminating unstable gradient computations while revealing progressive feature interactions that align with clinical reasoning. We demonstrate three foundational contributions: (1) normalized dot products generate anatomically meaningful explanations without backpropagation; (2) multi-component weighting captures multi-modal interactions in prompt-based architectures; and (3) layer-wise progression shows how models build reasoning from context to localized organ regions, enabling direct verification against medical expertise [Bhati et al., 2024, Gipiškis et al., 2024].

The limitations include dependence on ground-truth bounding boxes for prototype extraction, while Future work should explore unsupervised prototype learning and extend validation across diverse pathological conditions, demographic groups, and scanner manufacturers. This work establishes a foundation for clinically deployable explainability in medical image segmentation.

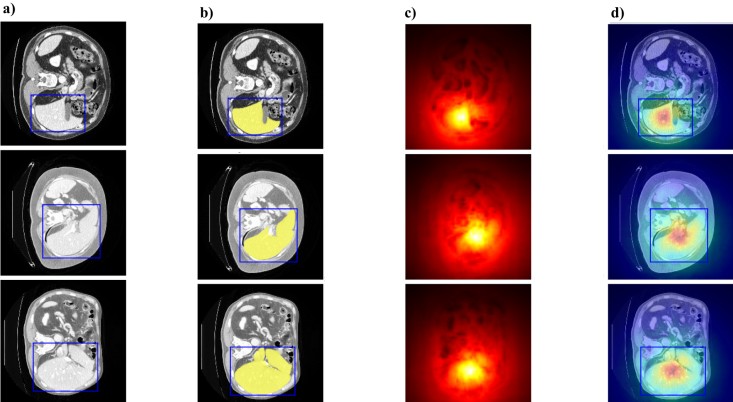

Figure 2: Sample output for 'Liver' a) input image b) segmented organ c) feature interaction d) Multi-level explainability

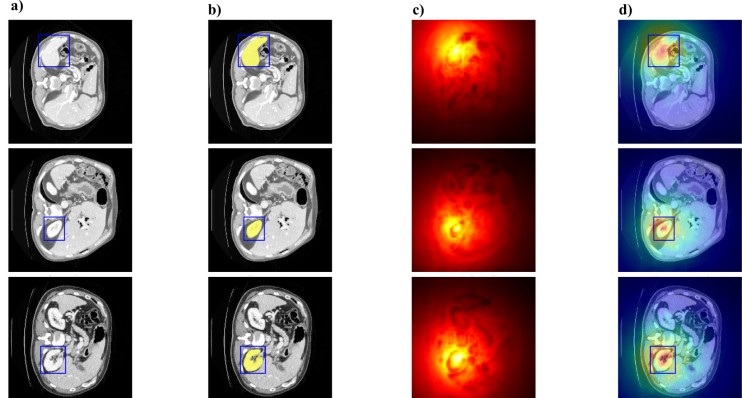

Figure 3: Sample output for 'spleen' a) input image b) segmented organ c) feature interaction d) Multi-level explainability

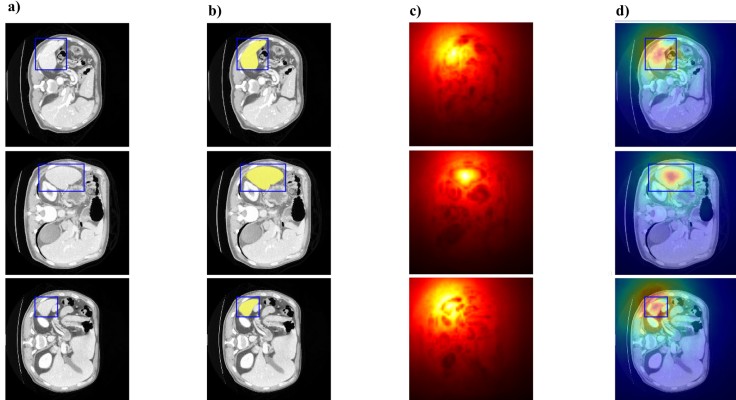

Figure 4: Sample output for 'Left Kidney' a) input image b) segmented organ c) feature interaction d) Multi-level explainability

## 3.1 Potential Negative Societal Impact

A key risk to this work is the misinterpretation of these explanations as proof of accuracy. The generated similarity maps reveal which regions influenced the prediction, but they do not measure how well the segmentation was performed. Overinterpreting visually coherent explanations can foster misplaced trust in flawed or biased models. The framework could also be misused to justify poor-performing systems by selectively presenting convincing maps, creating a false sense of reliability. Therefore, clear communication of its limits is essential. This tool should be used to aid the inspection of model reasoning, not as evidence of performance quality.

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
