# OpenReview forum: "Visual Semantics Meets Medical Diagnosis: Cross-Scale Embedding Alignment for Clinically Explainable Medical Image Segmentation"
_EurIPS.cc/2025/Workshop/MedEurIPS — EurIPS 2025 Workshop MedEurIPS Submission_

### Official Review · Reviewer_UQQw · 2025-10-31
**Novel, gradient-free explainability framework with promising contribution to clinical AI**

**Rating:** 9
**Confidence:** 4

**Review:**

This paper proposes a gradient‑free framework to generate anatomically grounded saliency maps for medical image segmentation models, aiming to improve explainable AI for clinical use.
### Pros
1. The paper introduces an innovative, gradient-free approach for explainable AI (XAI) in medical imaging. This method avoids the computational instability and vanishing gradient problems often seen in gradient-based techniques.
2. This framework tackles the critical issue of the "black box" problem in medical AI. the layer-wise analysis t shows how the model builds its reasoning from a broad contextual understanding to precise, localized anatomical regions.
3. The method validate its effectiveness on two distinct and large-scale public datasets, CHAOS and FLARE22, which together cover 13 abdominal organs.
### Cons
1. While the paper claims its explanations are more stable and localized than those from Grad-CAM, it will be better to include the results of Grad-CAM as well.
2. A significant limitation is that the current method relies on ground-truth bounding boxes to extract organ prototypes for its analysis. This dependency could limit its practical application in real-world scenarios where such clean, labeled data is not readily available.
3. Discussion on wether weakly/semi‑supervised variants could mitigate this constraint would improve practical applicability.

---

### Official Review · Reviewer_pBd3 · 2025-10-31

**Rating:** 6
**Confidence:** 3

**Review:**

The authors focus on generating XAI saliency maps for modern prompt-based segmentation models such as MedSAM. Overall, the paper is well written and well motivated. Standard saliency methods are either gradient-based or cannot be directly applied to modern vision-language models. The authors present an interesting new idea to produce clinically meaningful and anatomically relevant saliency maps, showing a progressive refinement of model focus across layers.

The authors also discuss key limitations of their approach, which relies on ground-truth bounding boxes as prompts, and outline potential directions for improvement. I believe that a quantitative evaluation and a larger qualitative study would be needed to draw any conclusions for the proposed approach. Importantly, the authors also thoughtfully consider potential social risks, such as inflated trust or a false sense of reliability that saliency-based methods may introduce. The paper presents an interesting new idea, and I believe it would stimulate valuable discussion at the MedEurIPS workshop.

---

### Decision · Program_Chairs · 2025-10-31

**Decision:**

Accept (Oral)

**Comment:**

Both reviewers commend the paper’s originality and relevance, noting that it introduces a novel, gradient-free explainability framework producing anatomically grounded saliency maps for medical segmentation.